# Damping Asymmetry Trimming Based on the Resistance Heat Dissipation for Coriolis Vibratory Gyroscope in Whole-Angle Mode

**DOI:** 10.3390/mi11100945

**Published:** 2020-10-19

**Authors:** Kechen Guo, Yulie Wu, Yongmeng Zhang, Jiangkun Sun, Dingbang Xiao, Xuezhong Wu

**Affiliations:** College of Intelligence Science and Engineering, National University of Defense Technology, Changsha 410073, China; guokechen@nudt.edu.cn (K.G.); ylwu@nudt.edu.cn (Y.W.); sunjiangkun15@nudt.edu.cn (J.S.); dingbangxiao@nudt.edu.cn (D.X.); xzwu@nudt.edu.cn (X.W.)

**Keywords:** whole-angle mode, hemispherical resonator gyroscope, damp tuning, resistance heat dissipation, damping asymmetry

## Abstract

Damping asymmetry is one of the most important factors that determines the performance of Coriolis Vibratory Gyroscope. In this paper, a novel damping tuning method for the resonator with parallel plate capacitors is presented. This damping tuning method is based on resistance heat dissipation and the tuning effect is characterized by the control force in Whole-Angle mode. As the damping tuning and stiffness tuning in the resonator with parallel plate capacitors are coupled with each other, a corresponding tuning system is designed. To verify the tuning effects, experiments are conducted on a hemispherical resonator gyroscope with Whole-Angle mode. The damping tuning theories is demonstrated by the testing results and 87% of the damping asymmetry is reduced by this tuning method with a cost of 3% decaying time. Furthermore, the angle-dependent drift in rate measurement after tuning is only 15.6% of the one without tuning and the scale factor nonlinearity decreases from 5.49 ppm to 2.66 ppm. The method can be further applied on the damping tuning in other resonators with symmetrical structure.

## 1. Introduction

Coriolis vibratory gyroscopes with integrated Micro-Electro Mechanical Systems (MEMS) are widely used in automotive, consumer electronics, industrial, aerospace, and other fields due to their small size, low power consumption, and low cost [1,2,3]. Conventional vibratory gyroscope works in Force to Rebalanced (FTR) mode as a rate gyroscope. In this mode, the gyroscope measures the angular rotation rate by forcing one of its resonance modes to vibrate and detect the Coriolis force-induced vibration of the second mode [4,5,6,7]. The most representative MEMS gyroscopes in FTR mode are the Disk Resonator Gyroscope (DRG) reported by Boeing [8] with a bias stability of 0.012°/h and the micro hemispherical resonator gyroscope reported by the University of Michigan [9] with a bias stability of 0.0103°/h. Although the FTR mode has excellent performance in rate measurement, it suffers from limited dynamic range and accumulative errors [10,11,12]. Whole-Angle (WA) mode is another control strategy of a vibratory gyroscope, in which the vibration pattern of resonator is allowed to freely precess along with the input of angular rate [13]. Compared to FTR mode, WA mode has the advantages of direct angular output, stable scale factor, and wide dynamic range [14].

However, WA mode has a strict requirement for the symmetry of a resonator and the main source of error in WA mode is the unequal mass, stiffness, and damping of the gyroscope’s two vibration modes [15]. Generally, the errors caused by stiffness asymmetry is handled by electrostatic frequency tuning [16,17,18] but the regulation of damping distribution is more sophisticated than the one of stiffness, which makes the damping mismatch still a non-negligible error source [19].

Damping in the resonators mainly comes from the air damping, surface loss, support loss, thermoelastic damping, and electrical damping [20,21,22]. And once the gyroscope resonator is vacuum packaged, only the electrical damping is tunable. In [23,24], the damping asymmetry is characterized by releasing the angle from an initial set point and observing the exponential decaying time constant. Though this approach is not suitable for gyroscopes of high quality, it provides the information for damping distribution. In [25], a weight energy control algorithm is proposed to keep the amplitude of oscillation constant in an inertial frame by maintaining the prescribed total energy. And in [26,27], an additional damping compensation loop is created based on the energy loop and reduces the root mean square of drift by 25%. Besides, virtual rotation is applied to average the error at a different pattern caused by damping asymmetry [28,29] and an offline extended Kalman filtering is proposed based on parameter identification and drift compensation for a MEMS ring vibratory gyroscope, reducing the angular drift to 1°/s [30,31]. Among these methods, only the effects caused by damping asymmetry are compensated but the physical damping asymmetry have not been reduced. The practical damping tuning method based on the resistance heat dissipation is presented and the method is verified by applying it on a comb-like single-axis gyroscope [32,33]. The damping tuning in the disk resonator gyroscope using parallel plate capacitors for driving is presented in [34]. However, in this article, the damping tuning is based on FTR mode and the tuning effect is only discussed on a single axis but not around the gyroscope. Besides, the approach to assess the damping tuning effect in [34] is testing the time constant, which is really an inefficient task.

This paper proposed a damping tuning method based on the resistance heat dissipation and Whole-Angle mode. The tuning theories are derived into the two degree-of-freedom resonator and the corresponding tuning system is designed. Instead of measuring the time constant, we take advantage of the energy control force in Whole-Angle mode to characterize the damping distribution. Experiments are conducted using a hemispherical resonator gyroscope under Whole-Angle mode and during the tuning, the damping and frequency changes are characterized by the control force in Whole-Angle mode. The research conclusions of this paper are also of great significance to other kinds of gyroscopes.

## 2. The Basic Theory of Damping Tuning

Figure 1 illustrates the basic structure of the damping tuning method based on resistance heat dissipation. In this system, the surface of the resonator and electrode form the two parallel plates of the capacitor and when the resonator is oscillating, the mechanical displacement at the parallel capacitor will generate an electric current. As the current flows through the resistance, part of its energy will be converted to heat, which means that the original mechanical energy is lost. The amount of energy loss can be controlled by the resistance and voltage values. In this way, the damping of the system is tunable. Define the initial capacitive gap between the resonator and electrode as $d_{0}$ and the effective area of the capacitor as *A*. In the case that the effective area of the resonator is large enough when compared with the capacitive gap, the fringing field effect is neglected and the capacitance C(x) can be calculated as:(1)$$C\left(x\right)=\frac{\varepsilon \varepsilon_{0}A}{d_{0}+x}$$

*x* is the relative displacement of the resonator surface and the motion of the resonator can be modeled as:(2)$$m\ddot{x}+c\dot{x}+kx=-\frac{\partial }{\partial x}\left(-\frac{1}{2}C\left(x\right)V^{2}\right)$$
where *m* and *k* are the effective mass and stiffness of the resonator and *c* is the damping factor. *V* is the voltage on the tuning electrode and the relationship between the *V*, and the bias voltage $V_{b}$ can be described as: (3)$$\begin{array}{c} V=V_{b}-R\frac{d}{dt}\left(C(x)V\right)\\ =V_{b}-RV\frac{dC(x)}{dt}-R^{2}\frac{d^{2}\left(C(x)V\right)}{dt^{2}}C\left(x\right) \end{array}.$$

Neglect the infinitesimal fraction o(R), the actual voltage on the tuning electrode *V* can be obtained:(4)$$V=\frac{V_{b}}{1-\frac{R\varepsilon \varepsilon_{0}A}{{\left(d_{0}+x\right)}^{2}}\dot{x}}.$$

Substitute the (Equation 4) into (Equation 2), and use Taylor expansion for *x* and $\dot{x}$, we can obtain the simplified dynamic equation by ignoring the higher-order term:(5)$$m\ddot{x}+\left(c+3\alpha^{2}V_{b}^{2}R\right)\dot{x}+\left(k-\frac{\alpha V_{b}^{2}}{d_{0}}\right)x-\frac{12\alpha^{2}RV_{b}^{2}}{d_{0}}\dot{x}x=-\frac{\alpha V_{b}^{2}}{2}$$
where constant α is defined as $\alpha =\varepsilon \varepsilon_{0}A/d_{0}^{2}$.

It can be seen that the damping factor and the effective stiffness of the resonator are affected by the tuning resistance *R* and the bias voltage $V_{b}$:(6)$$\Delta c=3\alpha^{2}V_{b}^{2}R,\Delta k=-\alpha V_{b}^{2}/d_{0}.$$

And the bias voltage will affect both the damping and stiffness while the resistance mainly affect the damping. Besides, there is a coupling term $x\dot{x}$ in (Equation 5), which will lead to the coupling of the damping and stiffness of the resonator and result in damping and stiffness nonlinearity. The ratio of the coupling term and damping related term can be described as:(7)$$r_{c}=-\frac{x}{d_{0}}\cdot \frac{12\alpha^{2}V_{b}^{2}R}{c+3\alpha^{2}V_{b}^{2}R}=-\frac{x}{d_{0}}\cdot \frac{4\Delta c}{c+\Delta c}.$$

Usually, the motion of the electrode is much smaller than the capacitive gap and the tuning should not change the initial damping too much. Hence, this ratio satisfies $|r_{c}|<<1$ and the coupling term’s influence in damping is negligible. Similarly, the ratio of the coupling term and stiffness related term is described as:(8)$$r_{k}=-\frac{\frac{12\alpha^{2}RV_{b}^{2}}{d_{0}}\dot{x}}{k-\frac{\alpha V_{b}^{2}}{d_{0}}}=-\frac{12\alpha R\dot{x}\Delta k}{k+\Delta k}=-\frac{12\Delta k}{k+\Delta k}\cdot \frac{RC_{0}\dot{x}}{d_{0}}$$
where $C_{0}=\varepsilon \varepsilon_{0}A/d_{0}$ is the essential capacitance. When the resonator is oscillating with its resonant frequency ω, the maximum of $\dot{x}$ is about $\omega x_{m}$, where $x_{m}$ is the maximum movement of the electrode. And the value of $r_{k}$ satisfies:(9)$$\left|r_{k}\right|<\frac{12\Delta k}{k+\Delta k}\cdot \frac{x_{m}}{d_{0}}\cdot RC_{0}\omega .$$

When using the resistance heat dissipation method, the time constant of the system $RC_{0}$ should be smaller than the oscillation period 2π/ω to guarantee the energy dissipating entirely in each oscillation period. The tuning effect on stiffness should also keep Δk<<k and the maximum movement should satisfy $x_{m}<<d_{0}$. Thus, it can be concluded that the coupling term’s influence in stiffness is also slight and this term is usually ignored for the simplification of the tuning process.

However, the analysis above is based on a linear model, which is limited by the onset of nonlinearities. According to the nonlinear theory [35,36,37,38], nonlinearity will cause higher order terms in *x* such as $x^{2}$ and $x^{3}$, to result in extra terms for stiffness. The additional terms ${\dot{x}}^{3}$, $\dot{x}x^{2}$, and $x^{2}\dot{x}$, which are all on the same order as $x^{3}$, will result in nonlinear dissipative contributions and interfere the damping tuning. Hence, this damping tuning method will not be suitable for resonators working with a large amplitude.

## 3. Practical Damping Tuning Method for CVG

### 3.1. The Concept of Damping Tuning in CVG

Based on the theory in Section 2, the method is applied to a two degree-of-freedom resonator with a symmetrical structure. Figure 2 shows the schematic of damping tuning with multiple electrodes.

The direction with maximum damping around the resonator is defined as the major damping axis and the orthogonal direction, which has the minimum damping, and is defined as the minor damping axis. $c_{1}$ and $c_{2}$ are the damping factors of major and minor damping axis. $\theta_{\tau }$ is the angle between the major damping axis and the direction of the excitation electrode. From the circular symmetry, the damping factor along an arbitrary direction can be described as:(10)$$c\left(\varphi \right)=c_{1}\cos^{2}\left(\varphi -\theta_{\tau }\right)+c_{2}\sin^{2}\left(\varphi -\theta_{\tau }\right)+\sum \Delta c_{i}\cos^{2}\left(\varphi -\theta_{i}\right)$$
where $\theta_{i}$ indicates the direction of each tuning electrode and $\Delta c_{i}$ is the damping change caused by tuning. After tuning, the new angle of damping axis can be calculated by:(11)$$\frac{\partial c(\varphi )}{\partial \varphi }\left|{}_{\varphi ={\theta^{\prime }}_{\tau }}\right.=0.$$

The solution of (Equation 11) is:(12)$$\tan2\theta_{\tau }^{\prime }=\frac{\left(c_{1}-c_{2}\right)\sin2\theta_{\tau }+\sum \Delta c_{i}\sin2\theta_{i}}{\left(c_{1}-c_{2}\right)\cos2\theta_{\tau }+\sum \Delta c_{i}\cos2\theta_{i}}$$
and the new damping factor of the damping axis became:(13)$$c_{1}^{\prime }=c_{1}\cos^{2}\left(\theta_{\tau }^{\prime }-\theta_{\tau }\right)+c_{2}\sin^{2}\left(\theta_{\tau }^{\prime }-\theta_{\tau }\right)+\sum \Delta c_{i}\cos^{2}\left({\theta^{\prime }}_{\tau }-\theta_{i}\right)$$
(14)$$c_{2}^{\prime }=c_{1}\sin^{2}\left(\theta_{\tau }^{\prime }-\theta_{\tau }\right)+c_{2}\cos^{2}\left(\theta_{\tau }^{\prime }-\theta_{\tau }\right)+\sum \Delta c_{i}\sin^{2}\left({\theta^{\prime }}_{\tau }-\theta_{i}\right).$$

The aim of damping tuning is to achieve $\theta_{\tau }^{\prime }=0$ and $c_{1}^{\prime }=c_{2}^{\prime }$, which can be described as:(15)$$\left\{\begin{array}{c} \left(c_{1}-c_{2}\right)\sin2\theta_{\tau }+\sum \Delta c_{i}\sin2\theta_{i}=0\\ \left(c_{1}-c_{2}\right)\cos2\theta_{\tau }+\sum \Delta c_{i}\cos2\theta_{i}=0 \end{array}\right..$$

Similarly, the theoretical tuning effect on stiffness can also be described as:(16)$$\left\{\begin{array}{c} \left(k_{1}-k_{2}\right)\sin2\theta_{\omega }+\sum \Delta k_{i}\sin2\theta_{i}=0\\ \left(k_{1}-k_{2}\right)\cos2\theta_{\omega }+\sum \Delta k_{i}\cos2\theta_{i}=0 \end{array}\right.$$
where $k_{1}$ and $k_{2}$ are the effective stiffness coefficients of the major stiffness axis and minor stiffness axis (the definition of the stiffness axis is similar to the one of damping axis). $\theta_{\omega }$ is the angle between the major damping axis and the direction of excitation electrode.

According to (Equation 5) and (Equation 6), the method simultaneously affect the stiffness and damping but the tuning effect on stiffness and damping is the opposite. Hence, if the angle between the stiffness axis and damping axis satisfies (Equation 17) and (Equation 18), it is possible to eliminate both the stiffness and damping asymmetry by this method:(17)$$\left(c_{1}-c_{2}\right)\left(k_{1}-k_{2}\right)\cos2\theta_{\tau }\cos2\theta_{\omega }\leq 0$$
(18)$$\left(c_{1}-c_{2}\right)\left(k_{1}-k_{2}\right)\sin2\theta_{\tau }\sin2\theta_{\omega }\leq 0$$

In this case, the bias voltage $V_{b}$ is adjusted to meet the requirement of frequency tuning and after the determining of bias voltage, the resistance can still be adjusted to complete the damping. However, if the distribution of damping cannot satisfied (Equation 17) or (Equation 18), the tuning of the frequency will intensify the damping asymmetry. On such occasions, the bias voltage is not determined by the frequency asymmetry and an additional frequency tuning system is needed to offset the tuning effect caused during the damping tuning as well as tune the original frequency asymmetry:(19)$$\left\{\begin{array}{c} \left(k_{1}-k_{2}\right)\sin2\theta_{\tau }+\sum \Delta k_{i}\sin2\theta_{i}+\sum \Delta k_{j}\sin2\theta_{j}=0\\ \left(k_{1}-k_{2}\right)\cos2\theta_{\tau }+\sum \Delta k_{i}\cos2\theta_{i}+\sum \Delta k_{j}\cos2\theta_{j}=0 \end{array}\right.$$
where $\theta_{j}$ and $\Delta k_{j}$ is the tuning parameters of the additional system. Theoretically, the bias voltage of the damping tuning in this case should be as small as possible to decrease the frequency tuning effect caused by itself.

Generally, the voltage adjustment is much more convenient than the resistance adjustment and the damping axis and stiffness axis can not always meet the requirements in (Equation 17) and (Equation 18), so it is better to divide the tuning method into two systems, one of which focuses on the damping tuning and ignores the side effect caused on stiffness while the other one only tunes the stiffness. And in each tuning system, the electrodes used can usually be divided into the axis tuning electrode whose major tuning effect acts on $\theta_{\tau }$ or $\theta_{\omega }$, and mismatch tuning electrode whose major tuning effect is on the damping or stiffness coefficients.

Particularly, if the axis tuning electrode is at 45° or 135° and the mismatch tuning electrode is at 0° or 90°, the equation of damping tuning can be simplified as:(20)$$\left\{\begin{array}{c} \Delta c_{\tau -axis}=\left|\left(c_{1}-c_{2}\right)\sin2\theta_{\tau }\right|\\ \Delta c_{\tau -damp}=\left|\left(c_{1}-c_{2}\right)\cos2\theta_{\tau }\right| \end{array}\right..$$

Similarly, the simplified stiffness tuning can also be described as:(21)$$\left\{\begin{array}{c} \Delta k_{\omega -axis}+\Delta k_{\tau -axis}=-\left|\left(k_{1}-k_{2}\right)\sin2\theta_{\omega }\right|\\ \Delta k_{\omega -freq}+\Delta k_{\tau -freq}=-\left|\left(k_{1}-k_{2}\right)\cos2\theta_{\omega }\right| \end{array}\right..$$

Substitute (Equation 6) into (Equation 20) and (Equation 21), then the bias voltage and resistance are determined as:(22)RaxisVτ−axis2=(c1−c2)sin2θτ(c1−c2)sin2θτ3α23α2RdampVτ−damp2=(c1−c2)cos2θτ(c1−c2)cos2θτ3α23α2Vω−axis2+Vτ−axis2=d0(k1−k2)sin2θωd0(k1−k2)sin2θωααVω−freq2+Vτ−damp2=d0(k1−k2)cos2θωd0(k1−k2)cos2θωαα.

Equation (Equation 22) provides an approach to calculate the required bias voltage and resistance. However, in actual practice, because the precise value of some parameters is hard to know, the calculation results can only provide an approximate value for tuning.

### 3.2. Approach for Damping Asymmetry Observation

The traditional method to characterize the damping is testing the decaying time of the resonator. However, the result obtained in this way only indicates the damping of a certain vibration pattern. Furthermore, in order to get the distribution of damping, experiments needs to be repeated under a different vibration pattern. In Whole-Angle mode, the test of damping distribution becomes much easier. According to Lynch’s theory [39,40],the gyroscope’s motion in WA mode can be solved using the method of averaging, resulting in three first-order nonlinear differential equations using which the dynamics of energy, quadrature error, pattern angle, and phase can be studied:(23)$$\dot{E}=-\left[\frac{2}{\tau }+\Delta \left(\frac{1}{\tau }\right)\cos2\left(\theta -\theta_{\tau }\right)\right]E-\frac{\sqrt{E}}{\omega }f_{as}$$
(24)$$\dot{Q}=-\frac{2}{\tau }Q-\Delta \omega \sin2\left(\theta -\theta_{\omega }\right)E+\frac{\sqrt{E}}{\omega }f_{qc}$$
(25)$$\dot{\theta }=-\kappa \Omega +\frac{1}{2}\Delta \left(\frac{1}{\tau }\right)\sin2\left(\theta -\theta_{\tau }\right)+\frac{1}{2}\Delta \omega \cos2\left(\theta -\theta_{\omega }\right)\frac{Q}{E}-\frac{f_{qs}}{2\omega \sqrt{E}}$$
(26)$$\dot{\delta \phi }=\dot{\phi }+\frac{1}{2}\Delta \omega \cos2\left(\theta -\theta_{\omega }\right)+\frac{1}{2}\Delta \left(\frac{1}{\tau }\right)\sin2\left(\theta -\theta_{\tau }\right)\frac{Q}{E}+\frac{f_{ac}}{2\omega \sqrt{E}}$$
where *E* and *Q* reflect the gyroscope’s energy and quadrature motion. θ stands for the vibration pattern angle and δϕ is the phase reference error. κ is the angle gain of the gyroscope. $f_{as}$, $f_{qc}$, $f_{qs}$, and $f_{qc}$ are the control force of each loop. 1/τ and Δ(1/τ) stands for the average and difference of the time constant while ω and Δω is the average frequency and frequency mismatch:(27)$$\begin{array}{c} \frac{1}{\tau }=\frac{c_{1}+c_{2}}{2m},\;\Delta \left(\frac{1}{\tau }\right)=\frac{c_{1}-c_{2}}{m}\\ \omega^{2}=\frac{k_{1}+k_{2}}{2m},\;\omega \Delta \omega =\frac{k_{1}-k_{2}}{m} \end{array}.$$

The method of averaging takes advantage of the fact that the resonator motion can be described as the behavior of a pair of coupled linear oscillators, both having natural frequencies very close to ω. The solutions will be very nearly sinusoidal oscillations at or near this frequency [39], with amplitudes and phases that vary slowly on a time scale determined by the period 2π/ω.

When the gyroscope is working, as shown in Figure 3, the control system should keep the resonator at the state of $\dot{E}\approx 0$, Q≈0, and $\dot{Q}\approx 0$. And the relative control force will become an efficient observer of the asymmetry: (28)$$f_{as}=-\omega \sqrt{E}\left[\frac{2}{\tau }+\Delta \left(\frac{1}{\tau }\right)\cos2\left(\theta -\theta_{\tau }\right)\right]$$
(29)$$f_{qc}=\omega \Delta \omega \sqrt{E}\sin2\left(\theta -\theta_{\omega }\right).$$

As shown in (Equation 24) and (Equation 25), the asymmetry of damping will cause the control force varying with the pattern angle of the gyroscope and sinusoidal portion of each force concerned with the corresponding mismatch and bias angle. As for $f_{as}$, there will even exist a bias of related with the average of damping factor, which can also be used to observe the damping changes.

## 4. Experimental Results

### 4.1. Device and Testing Platform

The experiments are carried out on the Hemispherical Resonator Gyroscope (HRG). The testing platform consists of two printed circuit boards. One is used to drive and sense the HRG’s vibration while the other is used for signal processing and the implementation of the controllers. The block diagram of this platform is illustrated in Figure 4. The digital hardware including phase-lock loop, modulation, demodulation, and control algorithms are all implemented in an Xlinx field-programmable gate array (FPGA) using Verilog programming language with a reference clock of 100 MHz. The 16 bit digital to analog converters(DAC) and analog to digital converters(ADC) have a sampling rate of 2 MHz. The experiment setup and the gyroscope are shown in Figure 5.

The gyroscope works in n = 2 mode and the resonant frequency of this mode is about 4.8 kHz. The resonator has 8 parallel plate capacitive electrodes inside that are used for sensing and driving in various control loops. Around the resonator, there exists 16 electrodes designed for tuning and the two opposite electrodes works as a pair. As shown in Figure 6, among these electrodes, 4 pairs of them can be used for the axis tuning of damping or stiffness while the other 4 pairs work for mismatch tuning. In addition, the electrodes used for damping tuning has to link to a resistance before the connection of bias voltage. To decrease the bias voltage needed in the damping tuning and to reduce the side effect in stiffness tuning, the resistance is determined as 9.1 MΩ.

In n=2 mode, if the angle between two tuning electrodes is 90°, these two electrodes will have the same tuning direction, which means the voltage on these electrodes will synchronously reduce the asymmetry or synchronously aggravate it. During the axis tuning or mismatch tuning, if the stiffness tuning effect caused by damping is opposite to the required effect, all 4 pairs of electrodes can be used (2 pairs for damping and 2 pairs for stiffness) while if the effect is the same, only 2 pairs can be used and the other 2 pairs need to be unconnected. As for the gyroscope used in the experiment, the electrode at 67.5° is used for the damping axis tuning while the electrode at 157.5° is used for stiffness axis tuning. The damping mismatch tuning uses electrodes at 0° and 90° and the frequency tuning uses ones at 45° and 135°.

### 4.2. Damping Axis Tuning and Mismatch Tuning

Before the tuning of the damping, the frequency tuning is roughly conducted to reduce the error caused by the frequency asymmetry and also spare some margins for damping tuning. When the voltage $V_{\tau -axis}$ varies from 0 to 25 V, the value of the control force $f_{as}$ is shown in Figure 7. As illustrated in Figure 7a, the value of fas has a sinusoidal vibration with the pattern angle, which is consistent with (Equation 28). As the tuning voltage increases, the curve of $f_{as}$ has an obvious upper-right movement, which indicates the changes of $\theta_{\tau }$ and 1/τ. The variation of the force bias $f_{as-ave}$ and $\theta_{\tau }$ is shown in Figure 7b.

When the voltage is 20 V, the phase of the curve is about 45° and the peak-to-peak value is the smallest, which is considered the finish of the damping axis tuning. Then the voltage on 0° and 90° is added and the result is shown in Figure 8.

Consistent with the conclusion in Section 3, the voltages at 0° and 90° do not change the angle of damping axis but reduce the mismatch of the two axis. As the tuning effect, the curve of $f_{as}$ only moves upward and the peak-to-peak value gradually declines. However, when the voltage is larger than 13 V, the peak-to-peak value of $f_{as}$ begins to increase, which indicates that the tuning is overdone and the relationship between fas and θ is revised. As shown in Figure 9a, the tendency of the curve $V_{\tau -damp}=15$ V is just opposite to other curves.

During the damping tuning, the frequency is also affected by the tuning voltage. Figure 9 shows the corresponding changes of $f_{qc}$ when the voltage is adjusted. As illustrated in the figure, the tuning effect is obvious and the tendency does agree with the theoretical analysis. As the frequency tuning will not affect the damping of the gyroscope, the remaining frequency asymmetry can be eliminated easily while the effect of damping tuning can be preserved.

Experiments are also conducted under a different environmental temperature, which affects the frequency and damping of the gyroscope. As shown in Figure 10a, when the temperature varies from 15 °C to 35 °C, the curve of the control force $f_{as}$ only moves upward but has little changes on its shape. Hence, it can be concluded that the temperature equally influences the damping all around the resonator and its effect on asymmetry can be almost neglected. Figure 10b shows the distribution of $f_{as}$ after damping tuning and the tuning voltages is fixed. As illustrated, in each temperature point, the tuning method is able to prevent $f_{as}$ from varying with pattern angle, which proves its the temperature robustness.

### 4.3. The Improvement after Damping Asymmetry Trimming

After more meticulous adjustments of the voltage, $V_{\tau -axis}$ is determined as 19V while $V_{\tau -damp}$ is determined as 11.8V and the reduction for damping asymmetry is shown in Figure 11a. When there is no tuning effect on the gyroscope, the frequency asymmetry interference the process of mode matching and the value of $f_{as}$ do not agree with the conclusion in (Equation 23). After frequency tuning, the curve is close to a standard sine curve and the peak-to-peak value of $f_{as}$ is about 113. When the damping tuning is added, the curve moves upward and is substantially flatter. The peak-to-peak value of $f_{as}$ is reduced to 14 and the bias of $f_{as}$ changes from 2701 to 2774, which indicates that the damping mismatch is reduced by 87% while the lost of decaying time is less than 3%. Figure 11b shows the measurement of the gyroscope under a fixed input rate. According to (Equation 26), the drift of precession rate is related to both the frequency and damping asymmetry. When without tuning, the drift is mainly caused by frequency asymmetry and the peak-to-peak value of measured rate is 0.0278°/s. After frequency tuning, there still remains a fluctuation with a peak-to-peak value of 0.0032°/s, which is considered the results of damping asymmetry. After the damping tuning, the residual angle-dependent drift is only 0.0005°/s, 15.6% of the one without damping tuning. The degree of improvement is close to the one observed from the control force $f_{as}$.

Since the damping tuning introduces additional losses into the system and reduces the quality factor, theoretically it will impact the sensitivity of the gyroscope to some degree. Figure 12a shows the result of the resolution test. As illustrated, the gyroscope after tuning can still sense the angular input of 0.001°/s which is the minimum output of our turntable. Figure 12b shows the Allan variance of the gyroscope. The angle random work before and after tuning is 0.019°/$\sqrt{h}$ and 0.021°/$\sqrt{h}$, which indicates the decrease of Signal-to-Noise Ratio after additional energy losses. As for the bias stability, it decreases from 0.73°/h to 0.68°/h after tuning. As the tuning effect mainly focuses on the symmetry of the gyroscope and losses of quality factor is only 3%, the influence in gyro’s static performance is not obvious. Besides, the test of scale factor is also conducted and the scale factor nonlinearity is shown in Figure 13. When there is no damping tuning, the nonlinearity is extremely severe at a small angular input due to the angle-dependent bias shown in Figure 11b. After tuning, the nonlinearity decrease from 5.49 ppm to 2.66 ppm, which reflects the effectiveness of reducing the asymmetry of gyroscope.

## 5. Discussion

According to the experimental results, the damping tuning method can efficiently reduce the asymmetry of damping and its side effect on stiffness tuning can be easily avoided by using an additional stiffness tuning system. Besides, this method has excellent temperature robustness which makes the tuning method insensitive to environmental changes. Though this method introduces additional energy dissipation, the losses in quality factor and resolution are acceptable and the improvement in gyroscope’s scale factor is obvious. Compared with the other compensation method dealing with the damping asymmetry, this method changes the essential damping distribution, which reduces the error caused by damping from its function principle. However, the current damping tuning method is only suitable for the linear system and there are still lots of neglected influencing factors such as fringing field effect, geometry, or external nonlinearities, which may be a topic in our further study.

## 6. Conclusions

This paper presented a damping tuning method based on the resistance heat dissipation. The tuning theories were derived and the corresponding tuning system was designed. Experiments were conducted using a hemispherical resonator gyroscope under Whole-Angle mode. During the tuning, the energy and quadrature control force were worked as the observer to characterize tuning effects on damping and frequency. The experimental results coincided with the theories and the method reduced 87% of the damping asymmetry with a cost of 3% decaying time. In addition, the angle-dependent drift in the rate measurement was also reduced and remained only 15.6% of the one without damping tuning, which led to the reduction of scale factor nonlinearity from 5.49 ppm to 2.66 ppm. The method presented in this paper is widely applicable to the other kind of gyroscope, which may be useful for further researches on inertial sensors with high accuracy.

## Figures and Tables

**Figure 1 micromachines-11-00945-f001:**
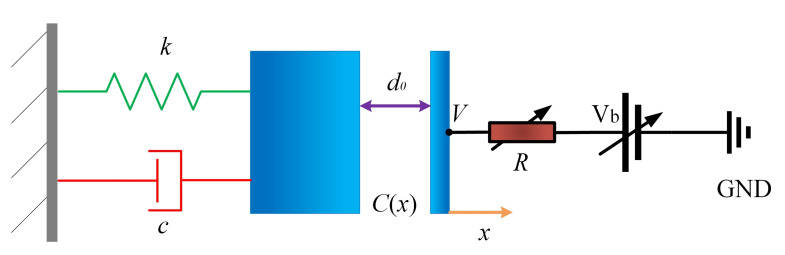
Schematic of the damping tuning based on the resistance heat dissipation.

**Figure 2 micromachines-11-00945-f002:**
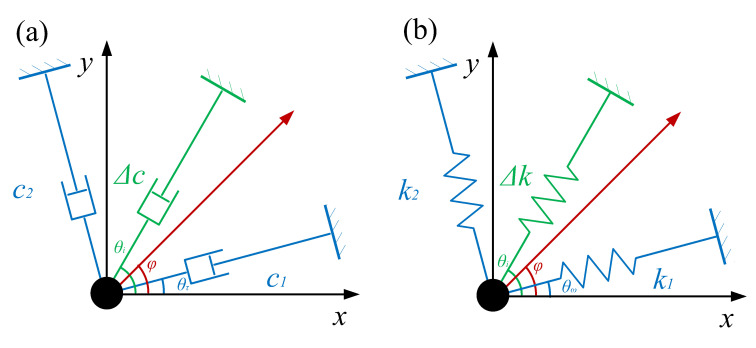
Damping tuning in a two degree-of-freedom resonator. (**a**) Effect on damping and (**b**) effect on stiffness.

**Figure 3 micromachines-11-00945-f003:**
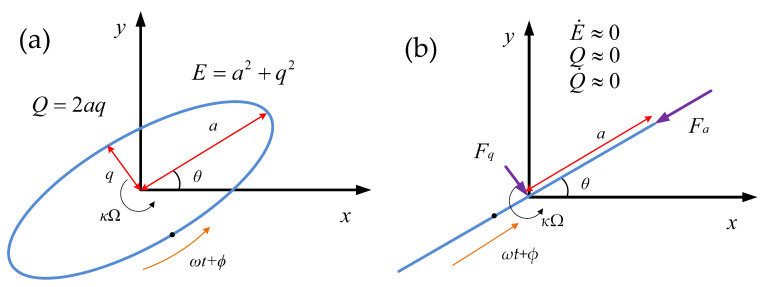
Illustration of the vibration pattern of an gyroscope in Whole-Angle mode. (**a**) The general vibration state and (**b**) the vibration state under control.

**Figure 4 micromachines-11-00945-f004:**
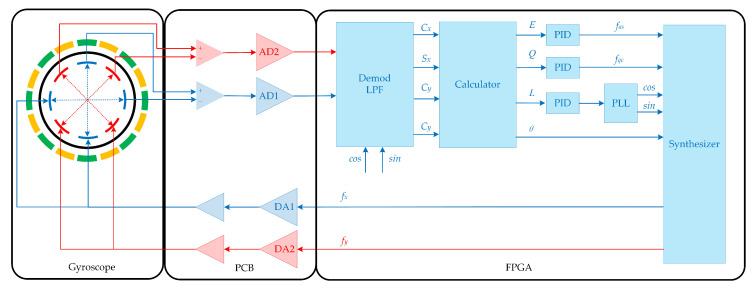
Block diagram of the electronics and control loops. The control system is implemented based on FPGA algorithm.

**Figure 5 micromachines-11-00945-f005:**
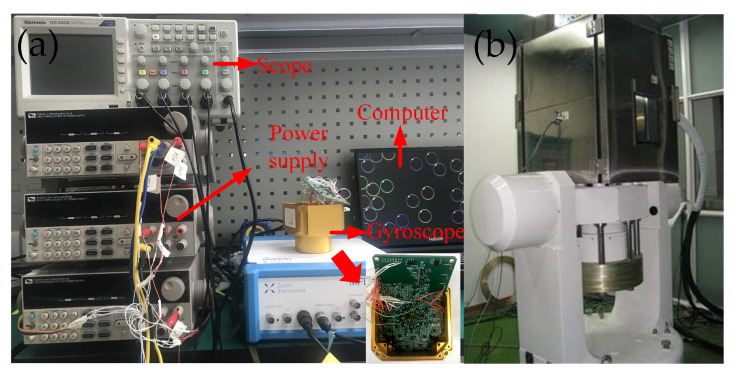
Experiment setup and the gyroscope for test.

**Figure 6 micromachines-11-00945-f006:**
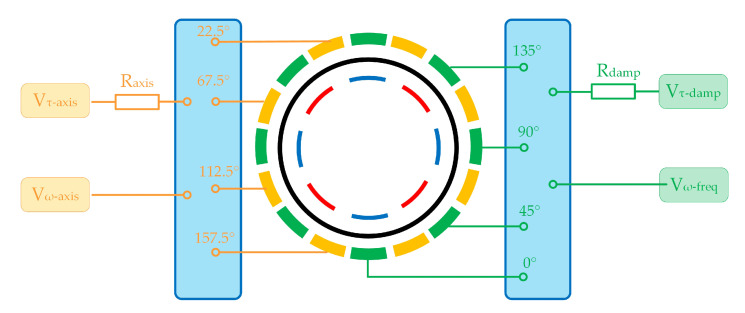
The schematic diagram of the connection between the tuning electronics and voltage sources.

**Figure 7 micromachines-11-00945-f007:**
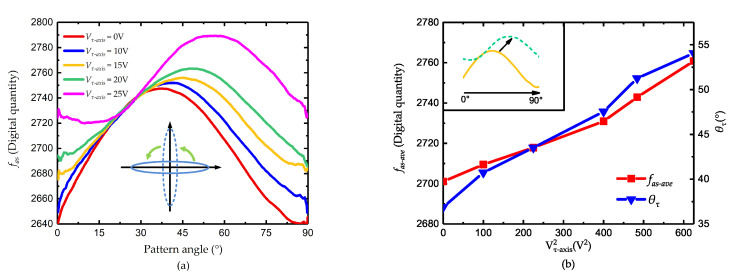
Damping axis tuning (**a**) the value of $f_{as}$, and (**b**) the value of $f_{as-ave}$ and $\theta_{\tau }$.

**Figure 8 micromachines-11-00945-f008:**
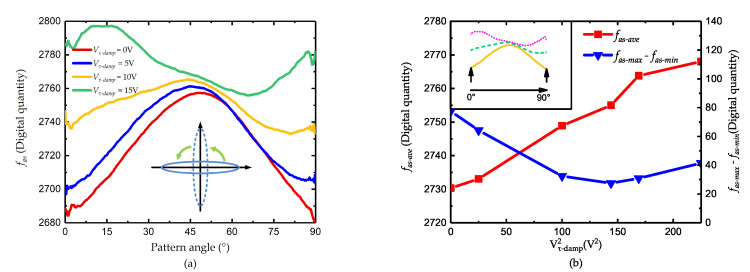
Damping mismatch tuning (**a**) the value of $f_{as}$ and (**b**) the value of $f_{as-ave}$ and $f_{as-max}-f_{as-min}$.

**Figure 9 micromachines-11-00945-f009:**
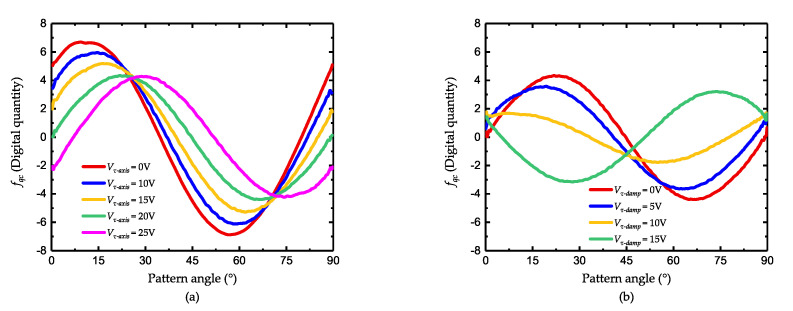
The value of $f_{qc}$ during damping tuning (**a**) during axis tuning and (**b**) during mismatch tuning.

**Figure 10 micromachines-11-00945-f010:**
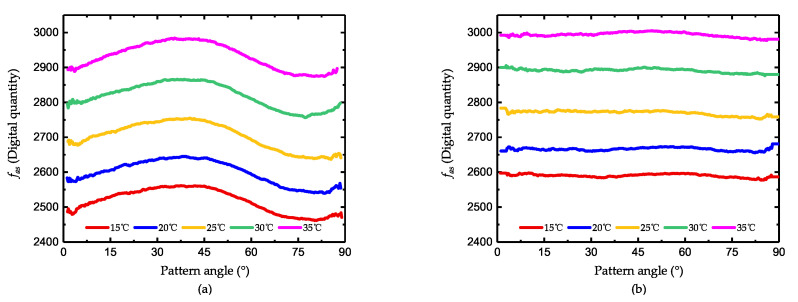
The temperature robustness experiment of damping tuning (**a**) before damping tuning and (**b**) after damping tuning.

**Figure 11 micromachines-11-00945-f011:**
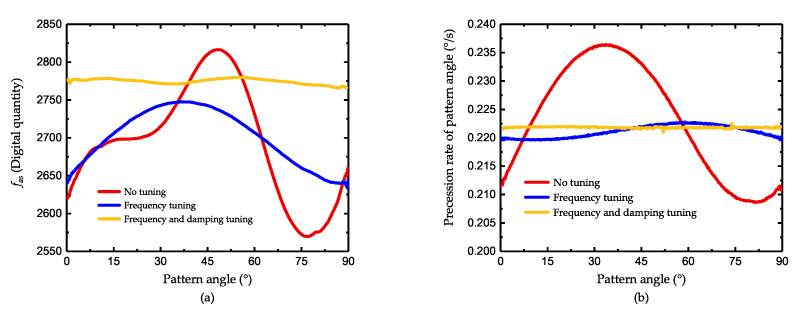
The improvement of gyroscope after tuning (**a**) the improvement in damping asymmetry and (**b**) the improvement in rate measurement.

**Figure 12 micromachines-11-00945-f012:**
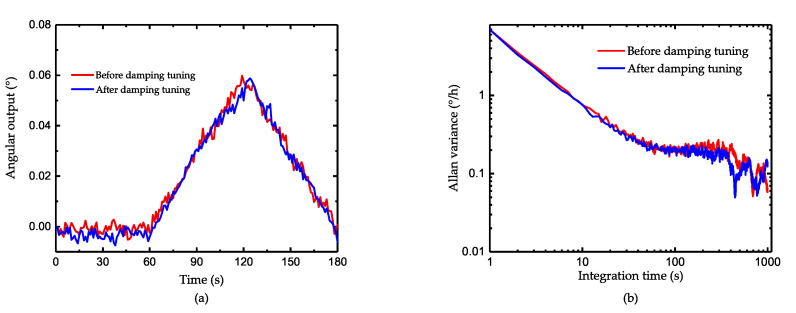
The test of gyroscope performance (**a**) resolution test and (**b**) Allan variance.

**Figure 13 micromachines-11-00945-f013:**
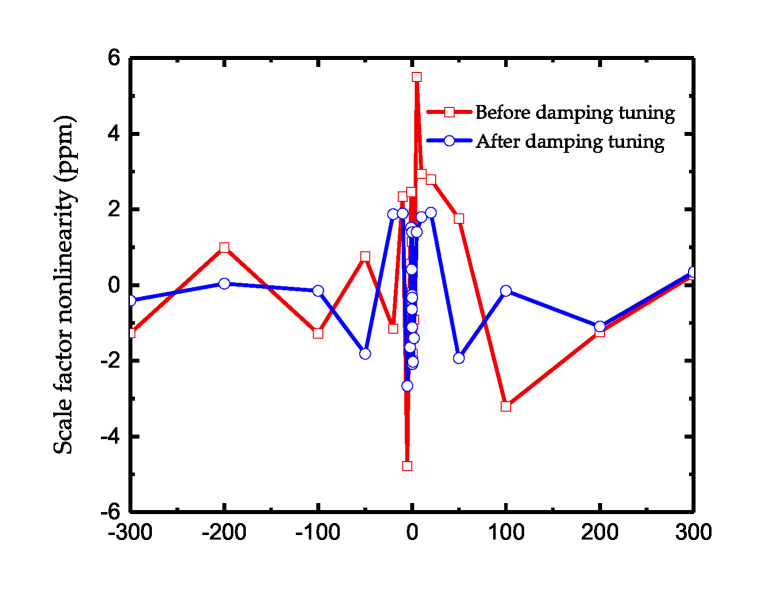
The scale factor nonlinearity of gyroscope before and after damping tuning.

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
