# Peer review of "Damping Asymmetry Trimming Based on the Resistance Heat Dissipation for Coriolis Vibratory Gyroscope in Whole-Angle Mode"

_micromachines, 2020, doi:10.3390/mi11100945_

Round 1
Reviewer 1 Report
The manuscript describes a method for minimizing angular stiffness and damping variations of a hemispherical gyroscope by means of adjustments applied to control electrodes. The principle relies on purposely introducing resistive losses to increase damping up to a point controled through the application of a bias voltage. Since this bias voltage also modifies the effective stiffness of the resonator in the vicinity of the electrodes, the authors describe a methodology to equalize both stiffness and damping over the whole angular range. They present an experimental demonstration to illustrate their method.
From a technical point of vue, the demonstration seems effective as far as one can guess, and the idea is sound. My only comment on this side would be that since the authors propose to introduce additional losses in the system to mitigate anisotropy, doesn't this impact the sensitivity or the resolution of the gyroscope ? Did they investigate such consequences of their approach ?
I would have more comments regarding the writting of the manuscript. The presentation of the author's works should be improved, as the manuscript is sometimes difficult to follow. To my opinion, this is in a large part caused by some clear definitions missing, or the use of some vague and imprecise wordings in some places:
1) Some acronyms are used but not defined: RMS (l. 37), DRG (l. 43)
2) In Figure 1, it would be good to introduce in the schematic where the voltage V is to be considered, and where the capacitance C is located.
3) In equation (3), a V is missing in the second line (term R dC(x)/dt should write R dC(x)/dt V).
4) Lines 55-57, the authors indicate that they neglect the term x dx/dt in equation (5) because it should be neglectible compared to the other terms. This seems however not so obvious, especially since the manuscript does not provide guesses for all terms, especially typical capacitance values, electrostatic gaps and displacement amplitudes. Could the authors add some numerical estimations to support this assumption ?
5) In section 3.1 and Figure 2, the authors introduce the wording "damping axes" or "axes of stiffness". At first read, this is really obscure and makes the following discussions hard to understand. Finally, I guessed that these axes correspond to the directions along which control electrodes are positionned along the periphery of the gyroscope. Is this correct ? In this case, I would highly recommend the authors to avoid trying to make their formulations compact, and to clearly formulate their ideas, even if this makes the text much longer. For example, they could start p. 4 with something like
[We] define c1 and c2 as the effective damping factor[s] for vibrations along the directions of the excitation electrodes, which are positionned at an angle θτ. From the circular symmetry, the damping factor along an arbitrary direction can be described as ...
Such clarifications would be welcome throughout the whole manuscript.
6) Similarly, could the authors clarify line 74 what is the meaning of "axis tuning electrodes" and "mismatch tuning electrodes" ? Do they refer to electrodes employed respectively to reach the conditions θ'τ=0 and c'1=c'2 mentionned line 63 ? Again, it would be good to state clearly the purpose of each electrode, even if this requires more sentences.
7) Line 113, the sentence "(...) if the angle between tuning electrode[s] is 90°, the tuning effect is the same." should be clarified. Which effect(s) did the authors have in mind ? And "the same" calls for a comparison between two things (two directions ?).
8) In sections 4.2 and 4.3, the authors rely heavily on the wording "performance", which again is vague and imprecise. l. 123, its occurence seems to refer to the value of the control force signal; l. 131 "As the performance" seems to mean "Quantitively"; l. 139 it seems to refer to "control value" or possibly to "efficiency" (clarification between the two would be welcome); l. 141 its occurence could be removed without prejudice, same for the occurence line 150.
Generally speaking, the quality of the English should be improved, if possible. Especially, running some grammatical checker would benefit to the manuscript.
Author Response
Dear Reviewer,
Thank you for your comments concerning our manuscript entitled “Damping asymmetry trimming based on the resistance heat dissipation for Coriolis Vibratory Gyroscope in Whole-Angle mode” (ID: micromachines-944014). Those comments are all valuable and very helpful for revising and improving our paper, as well as the important guiding significance to our researches. We have studied comments carefully and have made correction which we hope meet with approval. Revised portion are marked in yellow in the paper. The main corrections in the paper and the responds to the reviewer’s comments are as flowing:
Point 1: The demonstration seems effective as far as one can guess, and the idea is sound. My only comment on this side would be that since the authors propose to introduce additional losses in the system to mitigate anisotropy, doesn't this impact the sensitivity or the resolution of the gyroscope ? Did they investigate such consequences of their approach ?
Response 1: We are very sorry for our negligence that we didn’t provides the influence of damping tuning in the sensitivity and the resolution. The corresponding analysis and experiment is added from line 244 to 255. Theoretically, the additional energy losses will indeed weaken the sensitivity. However, because the losses of quality factor is only 3%, the impact in resolution is hard to figure out and the influence in gyroscope ‘s static performance is slight.
Point 2: Some acronyms are used but not defined such as RMS, DRG
Response 2: We have made correction according to your comments and the acronyms in paper are all explained before using.
Point 3: In Figure 1, it would be good to introduce in the schematic where the voltage V is to be considered, and where the capacitance C is located.
Response 3: We have replaced the previous figure into a new figure containing more information, which may help reader to understand the schematic.
Point 4: In equation (3), a V is missing in the second line (term R dC(x)/dt should write R dC(x)/dt V)
Response 4: We are very sorry for our incorrect writing. The equation has be corrected to the right form .
Point 5: Lines 55-57, the authors indicate that they neglect the term x dx/dt in equation (5) because it should be neglectable compared to the other terms. This seems however not so obvious, especially since the manuscript does not provide guesses for all terms, especially typical capacitance values, electrostatic gaps and displacement amplitudes. Could the authors add some numerical estimations to support this assumption ?
Response 5: Considering the your suggestion, we have added the analysis of the coupling term from line 85 to 104. The values of these terms are compared by evaluating their ratios and the neglection of the coupling may be more understandable.
Point 6: In section 3.1 and Figure 2, the authors introduce the wording "damping axes" or "axes of stiffness". At first read, this is really obscure and makes the following discussions hard to understand. Finally, I guessed that these axes correspond to the directions along which control electrodes are positioned along the periphery of the gyroscope. Is this correct ? In this case, I would highly recommend the authors to avoid trying to make their formulations compact, and to clearly formulate their ideas, even if this makes the text much longer. Similarly, could the authors clarify line 74 what is the meaning of "axis tuning electrodes" and "mismatch tuning electrodes" ? Do they refer to electrodes employed respectively to reach the conditions θ'τ=0 and c'1=c'2 mentioned line 63 ? Again, it would be good to state clearly the purpose of each electrode, even if this requires more sentences. Line 113, the sentence "(...) if the angle between tuning electrode[s] is 90°, the tuning effect is the same." should be clarified. Which effect(s) did the authors have in mind ? And "the same" calls for a comparison between two things (two directions ?).
Response 6: We have re-written this part according to your suggestion. We really feel sorry for our poor English which increased the difficulty of reading. We hope the expression can be easier for understanding after the revision.
Point 7: In sections 4.2 and 4.3, the authors rely heavily on the wording "performance", which again is vague and imprecise. l. 123, its occurrence seems to refer to the value of the control force signal; l. 131 "As the performance" seems to mean "Quantitively"; l. 139 it seems to refer to "control value" or possibly to "efficiency" (clarification between the two would be welcome); l. 141 its occurrence could be removed without prejudice, same for the occurrence line 150.
Response 2: We are very sorry for our lacks of vocabulary and incorrect expression. Most of the words mentioned in the comments are replaced by a more suitable one. We hope the revision can improve the reading experience.

Reviewer 2 Report
Excellent work excellent written, but with some shortcomings:
You are lacking an important reference:
DOI:10.1109/MEMS46641.2020.9056317
A Compact Microcontroller-Based MEMS Rate Integrating Gyroscope Module with Automatic Asymmetry Calibration
This obviously a journal paper of the conference paper [27]
Please check and include.
In Introduction: You should write what is the state of the art in performance, first of all drift in degree per second or hour.
I recommend to split the Chapter "Experiment results" into 2 chapters by making a separate chapter "Discussions" here you should also write to what extent your results are beyond state of the art justifying publication.
In line 98: Give reference [27] and the above paper.
Regarding Figure 4: Please include a photo of your experiment setup.
Regarding Figure 5: Maybe include a photo of the MEMS gyro.
Author Response
Dear Reviewer,
Thank you for your comments concerning our manuscript entitled “Damping asymmetry trimming based on the resistance heat dissipation for Coriolis Vibratory Gyroscope in Whole-Angle mode” (ID: micromachines-944014). Those comments are all valuable and very helpful for revising and improving our paper, as well as the important guiding significance to our researches. We have studied comments carefully and have made correction which we hope meet with approval. Revised portion are marked in yellow in the paper. The main corrections in the paper and the responds to the reviewer’s comments are as flowing:
Point 1: You are lacking an important reference: DOI:10.1109/MEMS46641.2020.9056317 Please check and include
Response 1: We are very sorry for our negligence that we didn’t include all relevant references. The reference is added into the introduction according to your suggestion. However, your comment that "In line 98: Give reference [27] and the above paper" really confuses us because the sentences in line 98 has no connection with the reference. Maybe there exists some mistakes about the line number. If there still need correction, please just let us know.
Point 2: You should write what is the state of the art in performance, first of all drift in degree per second or hour in Introduction.
Response 2: Considering the your suggestion, we have add more information about the topical gyroscope and their performance in introduction(line 21 to 24)
Point 2: I recommend to split the Chapter "Experiment results" into 2 chapters by making a separate chapter "Discussions" here you should also write to what extent your results are beyond state of the art justifying publication.
Response 2: We add the chapter "Discussions" according to your suggestion, in which we discuss the advantages and limitations of our method compared with other damping compensation methods. (line 257 to 266)
Point 3: Please include a photo of your experiment setup in Figure 5. Maybe include a photo of the MEMS gyro in Figure 5
Response 3: Considering your suggestion, we have add a new figure(Figure 5 in the revision), which includes the instrument, experiment setup and gyroscope.
Special thanks to you for your good comments.

Reviewer 3 Report
Report on paper "Damping asymmetry trimming based on the resistance heat dissipation for Coriolis Vibratory Gyroscope in Whole-Angle mode"submitted by Guo et al., for publication in Micromachines (micromachines-944014).
The authors proposed a damping tuning method based on the resistance heat dissipation and Whole-Angle mode. They derived the tuning theories for a two degree-of-freedom resonator and they designed the corresponding tuning system, which has been verified by conducting experiments on a hemispherical resonator gyroscope with Whole-Angle. Although the paper is interesting and the theoretical results are supported by experiments, the authors must perform some modifications by addressing the following comments:
- In the introduction, the literature survey lacks of references in the field of MEMS gyroscopes, which is a topic deeply investigated in the recent past.
- In the introduction, the authors should clearly highlight the originality of their paper with respect to their previous work in reference 28 where the same damping tuning method has been presented. Any overlap between the present work and reference 28 should be significantly reduced.
- In section 2, why the fringing field effect has been neglected without any justification according to the used assumptions?
- In section 2, the nonlinear coupling term has been neglected, while its effect was not evaluated especially if the resonator vibrates at large amplitudes. This point should be justified by the authors with respect to the dynamic range of the resonator in which, linearity is ensured.
- In subsection 3.2, the assumptions used to apply the averaging method should be clearly stated.
- The proposed model, which is linear, is limited by the onset of nonlinearities. The author should discuss that with respect to the recent literature related to vibrating MEMS operating in the nonlinear regime (for instance [Nonlinear Dynamics, 69, 1589–1610, 2012], [International Journal of Non-Linear Mechanics, 46(10), 1347-1355, (2011)], [Nonlinear Dynamics, 67, 859–883, 2012], [Appl. Phys. Lett. 117, 033502 (2020)]).
- What about the robustness of the proposed approach against temperature variations that may affect the sensor resonance frequencies?
- The authors could evaluate the performance enhancement of the gyroscope in term of resolution thanks to the proposed trimming approach.
Author Response
Dear Reviewer,
Thank you for your comments concerning our manuscript entitled “Damping asymmetry trimming based on the resistance heat dissipation for Coriolis Vibratory Gyroscope in Whole-Angle mode” (ID: micromachines-944014). Those comments are all valuable and very helpful for revising and improving our paper, as well as the important guiding significance to our researches. We have studied comments carefully and have made correction which we hope meet with approval. Revised portion are marked in yellow in the paper. The main corrections in the paper and the responds to the reviewer’s comments are as flowing:
Point 1: In the introduction, the literature survey lacks of references in the field of MEMS gyroscopes, which is a topic deeply investigated in the recent past.
Response 1: Considering the your suggestion, we have add more information about the typical MEMS gyroscope and their performance in introduction(line 21 to 24)
Point 2: In the introduction, the authors should clearly highlight the originality of their paper with respect to their previous work in reference 28 where the same damping tuning method has been presented. Any overlap between the present work and reference 28 should be significantly reduced.
Response 2: We have clarified the difference between this paper and the previous work according to your comments (line 49 to 52). The overlap only exists in the essential theory of the resistance heat dissipation.
Point 3: In section 2, why the fringing field effect has been neglected without any justification according to the used assumptions?
Response 3: We are very sorry for our negligence that we ignored the fringing field effect for granted. The assumption is added in line 69 and the influence of fringing field effect will be studied in our further research.
Point 4: In section 2, the nonlinear coupling term has been neglected, while its effect was not evaluated especially if the resonator vibrates at large amplitudes. This point should be justified by the authors with respect to the dynamic range of the resonator in which, linearity is ensured.
Response 4: Considering the your suggestion, we have added the analysis of the coupling term from line 85 to 98. The values of these terms are compared by evaluating their ratios and the neglection of the coupling may be more understandable.
Point 5: In subsection 3.2, the assumptions used to apply the averaging method should be clearly stated.
Response 5: We have added the assumption used in the averaging method according to your suggestion, which may help the readers to understand this simplification method
Point 6: The proposed model, which is linear, is limited by the onset of nonlinearities. The author should discuss that with respect to the recent literature related to vibrating MEMS operating in the nonlinear regime
Response 6: It is really true as you suggested that the model is limited by the nonlinearities. The influence of nonlinearities is roughly analysed from line 99 to 104 with respect to the suggested references. However, the current experiment can’t support the deeper analysis of nonlinearities. Maybe we will focus on the nonlinearities in further study.
Point 7: What about the robustness of the proposed approach against temperature variations that may affect the sensor resonance frequencies?
Response 7: Considering your suggestion, we have add experiments to test the temperature robustness. The result is presented from line 220 to 227. As illustrated in the experiments, the method has great temperature robustness when temperature varies from 15℃ to 35℃
Point 8: The authors could evaluate the performance enhancement of the gyroscope in term of resolution thanks to the proposed trimming approach.
Response 8: We have add the experiments relate with resolution and bias stability. Because the damping tuning mainly focus on the asymmetry trimming of the gyroscope, its influence in gyro’s static performance is slight according to the results. However, as for the scale factor nonlinearity, the improvement is obvious. The test result is presented in the revision from line 244 to line 255.
Special thanks to you for your good comments.

Round 2
Reviewer 3 Report
The authors have addressed my comments sufficiently to recommend publication of the paper in its current form.